# Google Earth Engine Sentinel-3 OLCI Level-1 Dataset Deviates from the Original Data: Causes and Consequences

Egor Prikaziuk *, Peiqi Yang and Christiaan van der Tol

Faculty of Geo-Information Science and Earth Observation (ITC), University of Twente,
7500 AE Enschede, The Netherlands; p.yang@utwente.nl (P.Y.); c.vandertol@utwente.nl (C.v.d.T.)
* Correspondence: e.prikaziuk@utwente.nl or prikaziuk@gmail.com; Tel.: +31-534-897-112

**Abstract:** In this study, we demonstrate that the Google Earth Engine (GEE) dataset of Sentinel-3 Ocean and Land Color Instrument (OLCI) level-1 deviates from the original Copernicus Open Access Data Hub Service (DHUS) data by 10–20 W m$^{-2}$ sr$^{-1}$ µm$^{-1}$ per pixel per band. We compared GEE and DHUS single pixel time series for the period from April 2016 to September 2020 and identified two sources of this discrepancy: the ground pixel position and reprojection. The ground pixel position of OLCI product can be determined in two ways: from geo-coordinates (DHUS) or from tie-point coordinates (GEE). We recommend using geo-coordinates for pixel extraction from the original data. When the Sentinel Application Platform (SNAP) Pixel Extraction Tool is used, an additional distance check has to be conducted to exclude pixels that lay further than 212 m from the point of interest. Even geo-coordinates-based pixel extraction requires the homogeneity of the target area at a 700 m diameter (49 ha) footprint (double of the pixel resolution). The GEE OLCI dataset can be safely used if the homogeneity assumption holds at 2700 m diameter (9-by-9 OLCI pixels) or if the uncertainty in the radiance of 10% is not critical for the application. Further analysis showed that the scaling factors reported in the GEE dataset description must not be used. Finally, observation geometry and meteorological data are not present in the GEE OLCI dataset, but they are crucial for most applications. Therefore, we propose to calculate angles and extraterrestrial solar fluxes and to use an alternative data source—the Copernicus Atmosphere Monitoring Service (CAMS) dataset—for meteodata.

**Keywords:** Google earth engine; GEE; Sentinel-3; ocean and land color instrument; OLCI; medium resolution imaging spectrometer; MERIS; Copernicus open access data hub service; DHUS; European Space Agency; ESA; pixel uncertainty; SNAP; time series; Copernicus atmosphere monitoring service; CAMS

## 1. Introduction

Many studies have been conducted on time-series of satellite images. Often a buffer of several pixels around the area of interest (AOI) is used. A typical workflow is the following: a user downloads all available images that contain AOI, extracts one pixel, and discards the rest. The Internet speed is usually a bottleneck of such an approach. If a user operates with European Space Agency (ESA) Sentinel data through Copernicus Open Access Data Hub Service (DHUS), the Long-Term Archive (LTA [1]) reduces the download speed further: a maximum of 2 simultaneous downloads per user, and a maximum of 20 requests per user per 24 h are permitted. In this way, a download of an image collection of 800 images would require 40 days. For popular satellites such as Sentinel-1 and Sentinel-2 alternative mirrors and data banks are available, such as Data and Information Access Services (DIAS [2]), which obtain their copy of data and do not have LTA-connected limitations.

In general, the approach described above—the "data-to-code" approach—does not seem efficient for collecting per-pixel time-series. The alternative solution is cloud computing, operating in the paradigm "code-to-data" or "moving code". The cloud computing

platforms vary by the level of required expertise: either users must be able to build their pipeline, specifying how CPU and memory resources of a cluster computer are used during the task execution, or a provider hides the realization of the storage, processing and infrastructure behind abstractions. The aforementioned DIAS system is an example of the former approach, whereas Google Earth Engine (GEE [3]) is an example of the latter. According to a recent comparison/review [4], GEE is the most user-friendly solution, although alternative platforms (OpenEO) provide more flexibility for scientists. Nonetheless, numerous scientific studies have been conducted in recent years with help of GEE [5,6]. If the data do not come from an official data provider, the conducted transformations, if any, have to be reported to ensure that the research is reproducible and independent from the chosen workflow ("data-to-code" or "code-to-data").

Our team has explored the theoretical applications of the recent Sentinel-3 [7] products for land surface monitoring using model simulations [8–10]. However, at the following proof-of-concept step using the "real-world" data [11], we faced the bottlenecks mentioned above—the preparation of a time-series dataset takes several days. Fortunately, in October 2017, GEE introduced a new Image Collection of Sentinel-3 level-1 Ocean and Land Color Instrument (OLCI) products. However, in July 2018 issues were reported [12]: the absence of angle (and meteorological) data bands, and the usage of tie-point instead of per-pixel geo-coordinates. These issues have not been resolved until now, although the Google Earth Engine developers group shows scientists' interest in it [13].

In this article, we demonstrate the challenges of an OLCI per pixel time series extraction (1), warn potential GEE OLCI dataset users about the hidden data modifications revealed during the comparison of per pixel time series between GEE and the official DHUS products (2), propose the method to augment the GEE OLCI dataset with angle and meteorological metadata (3), and propose a script for the Sentinel-3 data download avoiding LTA requests (4).

The paper is organized in the following way. The Materials and Methods section gives an overview of the Sentinel-3 OLCI product and its applications emphasizing time-series land monitoring. This is followed by the description of the pixel extraction workflow, GEE-DHUS name matching, and the proposed augmentation workflow. The Results and Discussion section describes the pixel positioning issues due to geo-coordinates or tie-point coordinates usage, pixel duplication and the absence of fixed tiles. This is followed by the pixel radiance differences between GEE and the official DHUS products, closing with the accuracy of the augmented data.

## 2. Materials and Methods

### 2.1. OLCI

#### 2.1.1. Products Overview

OLCI aboard Sentinel-3 (since 2016) [7] is the successor of Medium Resolution Imaging Spectrometer (MERIS [14]) aboard Envisat (operational from 2002 to 2012) [15]. The instruments provide observations in the visible–near-infrared domain from 0.4 to 1.0 µm (21 narrow bands OLCI; 15 bands MERIS) with pixel sizes of 300 m (full resolution (FR)) and 1200 m (reduced resolution (RR)). The land products are disseminated at two levels:

- level-1:
  - top of atmosphere (TOA) radiance per band
- level-2
  - integrated water vapour (IVW)
  - OLCI (MERIS) terrestrial chlorophyll index (OTCI, MTCI)
  - OLCI (MERIS) global vegetation index (OGVI, MGVI)
  - top of canopy (TOC) red (681 nm) and near-infrared (865 nm) reflectance

MERIS level-2 product contains more data: top of canopy (TOC) and top of atmosphere (TOA) reflectance in all bands, ocean and cloud products, some of which are available as individual Water and Synergy products of Sentinel-3, see Table A1 for details.

In the case of OLCI, RR products are distributed in the form of stripes (stretching from pole to pole) and FR products are disseminated as frames [16] (pieces of those stripes). Unlike fixed tiles of Sentinel-2 products, OLCI stripes and frames correspond to one of 365 orbits [17]. In this way, a single point can be viewed from up to 27 different orbits (according to the repeat cycle). Furthermore, pixel coordinates are reported per flight line (i.e., the grid is irregular or not orthorectified), which for time series analysis leads to gridding artifacts [18].

OLCI is currently onboard two satellites: Sentinel-3A (launched 16 February 2016) and Sentinel-3B (launched 25 April 2018). The employment of the constellation of Sentinels reduces the revisit time from 1.8 (1 spacecraft) to 0.9 (2 spacecrafts) days (Table 1 on [19]), yet it creates the need for a cross-calibration of the instruments. A unique 5-month cross-calibration of the instruments was conducted in the so-called "tandem phase" [20] from June to mid-October 2018 [21]. It is important to take this phase into account during time series preparation, especially the drift phase (mid-October–end-November) when Sentinel-3B satellite images were taken from non-nominal orbits that do not occur during the normal operational use.

In terms of data availability, there are two types of products—near-real-time (NR) and not-time-critical (NT). The products differ by meteodata but the radiance matches. NR products are available within 3 h after the acquisition and retained in the archive for up to 2 months. Finally, with the updates of algorithms the whole archive might be reprocessed, which is indicated closer to the end of the full product name as operational (O) or reprocessed (R) (Listing 1).

### 2.1.2. Time Series Applications for Land

The primary aim of both OLCI and MERIS instruments is the ocean color monitoring; however, land applications have also been largely explored, mostly using vegetation reflectance indices such as natural difference vegetation index (NDVI), MTCI, and MGVI. The MTCIproduct was used for phenology monitoring in tropical India [22,23], in the UK [24], and the whole of Europe [25]. Furthermore, Atkinson et al. [26] compared MTCI smoothing methods in relation to the phenological stage. The MGVI was used as a seasonal pattern indicator [27,28] or as a proxy of green instantaneous absorbed photosynthetically active radiation [29–31]. Level-1 data have been used for biophysical properties retrieval: leaf area index (LAI) [32–34] and leaf and canopy chlorophyll content [34,35]. Burned area mapping is another application of MERIS products [36–38]. Zurita-Milla et al. [39] proposed a MERIS pixel unmixing method for the resolution sharpening.

We found only three studies that used the time series of OLCI so far. Pastor-Guzman et al. [40] demonstrated that OTCI is a robust successor of MTCI: the seasonal cycles of both indices in various ecosystems were identical. Rather short (30 days) time series of level-1 data were used for burned area estimation with OLCI-derived NDVI [41]. Yang et al. [11] made a time-series LAI retrieval algorithm from TOA OLCI data. Single-time applications of single-image OLCI data were used for biophysical properties retrieval [42] (Synergy product) and for demonstration of radiance calibration network [43].

As was mentioned in the introduction, many OLCI studies were conducted on synthetic (either model-simulated or resampled) datasets: simulation of TOA signal [8], estimation of leaf chlorophyll content [44], canopy chlorophyll content [45] grass chlorophyll and nitrogen content [46], nitrogen–phosphorous ratio [47], LAI [48], and biochemical parameters [10]. Wang and Atkinson [49] proposed an algorithm of Sentinel-2 spatio-temporal gap-filling with OLCI data for red, green, blue, and near-infrared bands. A number of studies focused on the future ESA Earth Explorer 8 (FLEX) mission [50]: scene simulation [51] and sun-induced fluorescence retrieval [9].

Overall, OLCI (mostly MERIS) data have been widely used for land monitoring in time. In studies where mapping was not an objective of the study, a spatial smoothing of 3-by-3 pixel area has been used [32,40,52].

### 2.1.3. OLCI Level-1 Full Resolution Product

OL_1_EFR product contains TOA data for 21 bands, quality flags, and tie-point grid data. Data provided per band are reflected radiance (Oa*_radiance), solar flux (solar_flux_band_*), central wavelength (lambda0_band_*), and full width half maximum (FWHM_band_*). The coordinate set for these data is per pixel geo-coordinates (latitude, longitude, and altitude).

The quality flag is a 32-bit integer: bits from 0 to 20 indicate per band saturation, the remaining 11 bits provide additional information. Note that cloud flags are not present, but quality_flags_bright (bit 27) can be a proxy for cloud detection. The goal of this study was to compare all available images (not only cloud-free); thus, we did not use this flag. Another important quality flag is quality_flags_duplicated (bit 23). If the pixel is marked as duplicated (23% of any image) all per band data, including geo-coordinates, is taken from the nearest (and the closest to the track) neighboring pixel (Figure 1). Duplicated pixels are an unavoidable consequence of the swath width, satellite height, and the curvature of the Earth; off-nadir pixels are viewed from larger angles which makes the projected area (field of view, FOV) larger, while the product grid is equally spaced. Because tie-point coordinates and meteodata (see the next paragraph) are interpolated, they are not prone to duplication.

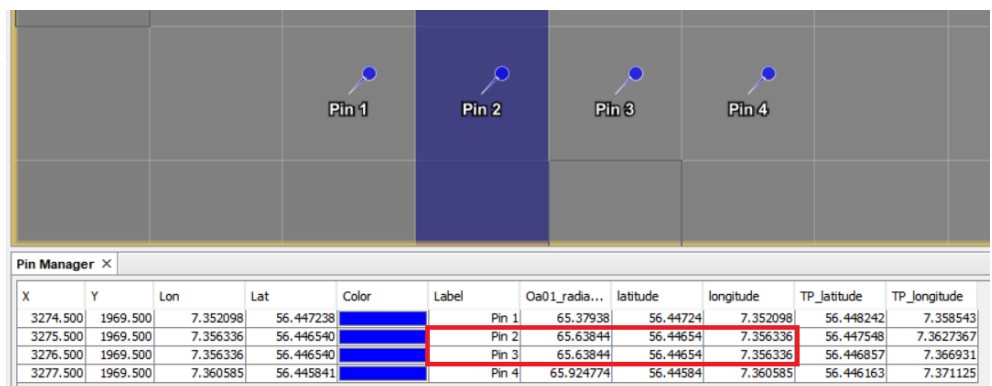

**Figure 1.** An OLCI image viewed with zoom in SNAP. Squares—single pixels, the blue stripe— quality_flags_duplicated mask. The pins demonstrate that the duplicated pixel (Pin 2) copies radiance and geo-coordinates (but not tie-point (TP) coordinates) from the nearest pixel (Pin 3). Duplicated pixels occur very often (23% of any OLCI image).

An additional component of the OLCI level-1 product is tie-point grid data, comprising sun–sensor–target geometry (solar and observation azimuth and zenith angles) and meteorological metadata coming from a European Centre for Medium-range Weather Forecasts (ECMWF) dataset: wind speed, humidity at 850 hPa, sea level pressure, atmospheric water vapor and ozone content, and atmospheric temperature profile. Tie-point data have different coordinates (TP_latitude, TP_longitude) and different resolution in across-track (along longitude) direction (77 columns for tie points versus 4865 columns for geo coordinates) resulting in 16 km spacing. Tie-points (TP) are a subsampled version of the image grid; as such, association between TP and pixels shall be done via their image coordinates (with the appropriate sub-sampling factor $(4865 − 1)/(77 − 1) = 64$ (for FR)) and not from geo-coordinates which are different. TP is the intersection of the line-of-sight from the satellite with the reference ellipsoid, which entails the absence of the observation parallax correction.

The Google Earth Engine "COPERNICUS/S3/OLCI" image collection [53] contains only reflected radiation bands, the quality flags band, and some metadata within image properties, including the official product name (PRODUCT_ID). Meteorological data and observation geometry, required for the atmospheric correction and further processing, are absent. As mentioned in the issues [12], GEE products use tie-point coordinates, not per pixel geo-coordinates. There are several data transformations present in the GEE

OLCI dataset. First, GEE reports per band scaling factors which are equal to those of the corresponding DHUS NetCDF per band radiance files. In the example on [53], after the scaling factor application visible radiance does not exceed 6 W m$^{-2}$ sr$^{-1}$ μm$^{-1}$, which is not realistic. From the range of provided radiance values (without scaling), we conclude that they are just the rounded original values. Therefore, we operated with GEE values without scaling and recommend the users to do so. Second, the images of the GEE collection have different sizes: the original product is always [4865, 4090], while the GEE version depends on the coordinate reference system (CRS), and often has the x-size of 4866, whereas the y-size ranges from 2895 to 4604. The number of pixels changes, as well as the ground resolution (from nominal 300 m to 318 m in GEE), but we were not able to find the interpolation technique that GEE used during reprojection.

### 2.2. Workflow

#### 2.2.1. Time Series Preparation

We downloaded all available 1987 not-time-critical (NT) products (from 26 April 2016 to 12 September 2020) of OLCI level-1 full resolution images containing Speulderbos eddy covariance site (NL-Spe, 52.251185°N, 5.690051°E, elevation 64 m). The choice of a single site located in the European area is a limitation of this study, which would benefit from a multi-site comparison across different continents and altitudes.

Out of 1987 products, 1109 online (not in LTA) images were downloaded from Copernicus Open Access Data Hub Service (DHUS [54]), the other 878 offline (in LTA) images were downloaded from the alternative mirror—Level-1 and Atmosphere Archive and Distribution System—Distributed Active Archive Center (LAADS DAAC [55]). Pixel time series were extracted with Sentinel Application Platform (SNAP [56]) version 8.0 software, Extract Pixel Values tool v1.3. Quality flags band was extracted incorrectly (probably, rounded due to the length of the integer), thus we extracted it with a custom script directly from the product's NetCDF file ("qualityFlags.nc") taking the pixel x, y coordinates suggested by the tool.

We quantified the uncertainty of four approaches to pixel extraction:

1. single-pixel extraction based on geo-coordinates (Figure 2b), SNAP;
2. single-pixel extraction based on tie-point coordinates (Figure 2c);
3. 3-by-3 pixel extraction based on geo-coordinates with further averaging; and
4. 3-by-3 pixel extraction based on tie-point coordinates with further averaging.

SNAP Extract Pixel Values tool conducts extraction exclusively based on geo-coordinates. This behavior does not depend on the reading option available in the Sentinel-3 Toolbox: "Read Sentinel-3 OLCI products with per-pixel geo-coding instead of using tie-points". Therefore, for the cases of tie-point-coordinates-based extraction (cases 2 and 4), we extracted a larger area (7-by-7 pixels) and calculated the nearest pixel based on the corresponding tie-point coordinates ourselves.

For the same period per pixel (NL-Spe, 52.251185°N, 5.690051°E) time series along with PRODUCT_ID property (full product name), latitude, longitude, coordinate reference system (CRS), pixel x and y coordinates, and image width and height were extracted from the Google Earth Engine "COPERNICUS/S3/OLCI" dataset [53]. Single pixel extracts were compared to DHUS single pixel extractions, the extractions with a 450 m radius mean region reducer (the equivalent of 3-by-3 pixels) were compared to the spatially aggregated DHUS extractions.

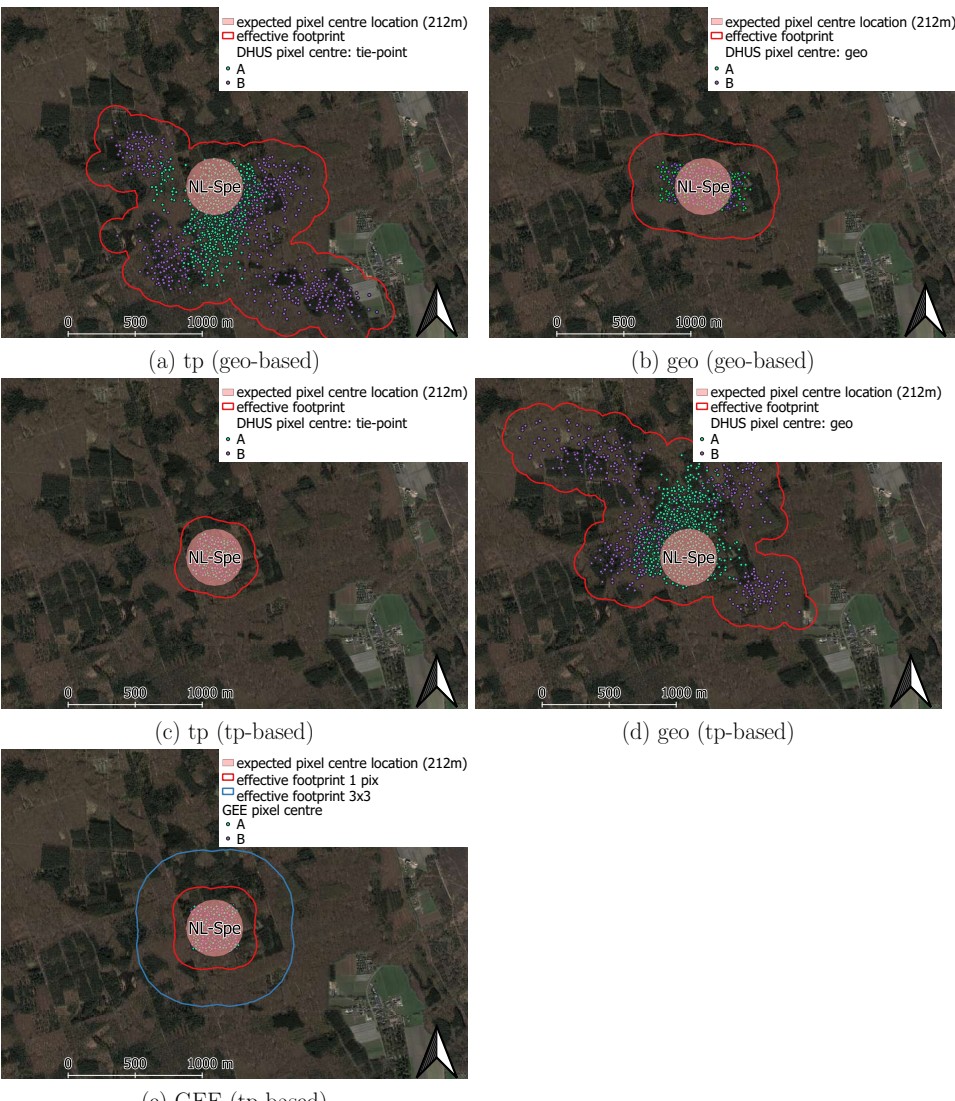

**Figure 2.** The effective ground footprint of Sentinel-3 OLCI dataset at Speulderbos site from DHUS images (red polygon). The pink circle is the 212 meter radius circle of the theoretically expected pixel center location. The points show the actual OLCI pixel center location of Sentinel-3A (green) and Sentinel-3B (purple) spacecrafts. The footprint was calculated based on (**a**) tie-point coordinates, corresponding to (**b**), (**b**) geo-coordinates (standard SNAP extraction), (**c**) tie-point coordinates (modified SNAP extraction), and (**d**) geo-coordinates corresponding to (**c**), (**e**) GEE coordinates (reported to be tie-point coordinates).

### 2.2.2. Distance Control

The extracted pixel coordinates were imported to QGIS [57] 3.4.14-Madeira, where a circular buffer of 212 m (radius of the circumscribed circle around a 300 m by 300 m square pixel, i.e., the expected pixel center location) was drawn (225 m for GEE, as GEE OLCI resolution is 318 m). The underlying map layer for visualization was Google Satellite from QGIS Server. The distances between coordinate points were calculated with the distance() function from the geopy package (v1.21.0) in Python 3.7.5.

### 2.2.3. GEE and DHUS Name Matching

GEE and DHUS extracted values were joined by the product name (either full or short) and radiance values were compared by a simple root mean square error (RMSE) metric and a relative RMSE (rRMSE)—the ratio of RMSE to the average band radiance.

For DHUS, out of 2002 products we were able to download 1987 and extract pixels from 1908. For GEE, the number of products in the filtered by geo-region collection was 6795. However, most of them were wrongly reprojected (infinite image boundary boxes) or did not have any assigned CRS, resulting in 2045 valid points. Some products (129 DHUS, 166 GEE) have no-data at their margins, leading to no radiance extracted. As long as that influenced neither matching nor metrics, we did not remove them. Overall, we matched 1146 images by the full name (PRODUCT_ID) and 1887 images by the short name ("system:index" in GEE terminology) (Table 1).

**Table 1.** Statistics on image comparison.

| DHUS | | Filter | GEE | |
|---|---|---|---|---|
| **Left** | **Dropped** | | **Left** | **Dropped** |
| 2002 | | products | 6795 | |
| 1987 | 15 | loaded/CRS present | 2221 | 4574 |
| 1908 | 79 | extracted/CRS valid | 2045 | 176 |
| 1146 | 762 | matched by full name | 1146 | 899 |
| 1887 | 21 | matched by short name | 1887 | 158 |

The short name does not uniquely identify a product: the second half of the name contains information about (re)processing time, the ground processing unit, and time dependency. Listing 1 shows two full names, corresponding to the same short name. Indeed, the images belong to the same acquisition (11 December 2018), still the first one was released on 12 December 2018 by Land OLCI Processing and Archiving Centre 1 (LN1) in operational (O) mode, whereas the second one was reprocessed (R) on January 2020 by Marine Reprocessing Centre 1 (MR1).

**Listing 1.** An example of two full names ("PRODUCT_ID" in GEE) that result in the same short name *S3B_20181211T093534_20181211T093710* ("system:index" in GEE). The difference in the reprocessing time and the reprocessing unit is highlighted in bold. The number in red is the relative orbit number used for viewing angles calculation.

```
S3B_OL_1_EFR____20181211T093534_20181211T093710_20181212T133634_0096_019_307_1980_LN1_O_NT_002
S3B_OL_1_EFR____20181211T093534_20181211T093710_20200115T181744_0096_019_307_1980_MR1_R_NT_002
```

Among the 762 DHUS images that were not matched by the full name, we managed to uniquely match 741 by the short name (Table 1). Those matched images include

- 211 GEE products that came from Svalbard Satellite Core Ground Station (SVL), which is not presented in DHUS;
- 300 (including 211 SVL) GEE products were from near-real-time (NR) dataset, whereas we took only non-time-critical from DHUS (NT);
- 218 GEE operational products (O), processed in 2019 by LN1, were reprocessed (R) in 2020 by MR1;
- 2 products mismatched by processing time

We did not find any SVL products in the collection apart from the ones from 2018, probably, GEE switched to the official DHUS archive. Finally, 336 GEE products did not have any full name (empty PRODUCT_ID property), yet 221 of them matched by the short name. As far as we could check, the absence of the full name is the case for the global GEE collection only in 2017 (from February to December 2017); currently all images can be successfully traced back by the full name.

### 2.3. GEE Augmentation

#### 2.3.1. Angles

The GEE OLCI product does not provide the sun–sensor–target geometry that is crucial for scientific applications. Solar angles can be calculated based on point coordinates and time. However, the calculation of observation angles is not that straightforward, due to the absence of fixed tiles. We propose to use the relative orbit number information (characters 74:77 of the official product name). As the repeat cycle is 27 days, it is enough to download a set of images with unique orbits, extract viewing geometry from them and use it for all other products from the same orbit. The orbits for the study site (NL-Spe) are presented in Table 2.

**Table 2.** Frequent orbits for the NL-Spe site.

| Orbit Number | Counts Full Name | Counts Short Name |
|:---:|:---:|:---:|
| 8 | 77 | 15 |
| 22 | 81 | 15 |
| 36 | 80 | 15 |
| 51 | 53 | 15 |
| 65 | 78 | 15 |
| 79 | 79 | 16 |
| 93 | 77 | 14 |
| 108 | 81 | 16 |
| 122 | 79 | 16 |
| 136 | 80 | 16 |
| 165 | 78 | 17 |
| 179 | 76 | 15 |
| 193 | 77 | 15 |
| 222 | 79 | 14 |
| 236 | 78 | 15 |
| 250 | 79 | 15 |
| 279 | 78 | 15 |
| 293 | 77 | 15 |
| 307 | 79 | 15 |
| 336 | 78 | 17 |
| 350 | 77 | 15 |
| 364 | 77 | 15 |

#### 2.3.2. Meteo

The GEE OLCI product does not have meteorological data, which is necessary information for applying the atmospheric correction. We downloaded the data from the near-real-time Copernicus Atmosphere Monitoring Service (CAMS [58]) dataset of ECMWF. The CAMS dataset provides variables at model levels, pressure levels (from 1 to 1000 hPa), and at surface level at $0.125 \times 0.125$ degree resolution. Air temperature profile (NetCDF variable name t, parameter ID 130) and relative humidity (r, 157) at 850 hPa were taken at pressure levels, the other parameters—total column water vapor (tcwv, 137), GEMS total column ozone (gtco3, 210206), mean sea level pressure (msl, 151), and 10 meter U and V wind components (horizontal wind vectors u10, v10)—at surface level. The dataset has a 3-hourly timestamp. We used the initialization time of 00:00 UTC and steps of 9 and 12 h. The value at the time of Sentinel-3 overpass was interpolated by time and geographical coordinates with interp() function of xarray package (v0.16.0) in Python 3.7.5. As a proof of concept we used January 2020 data for comparison.

Although the source of meteorological data is mentioned explicitly in the Product Data Format Specification of SLSTR level-1 products (Section 4.2.1.9 [59]), the specific ECMWF dataset used in the OLCI dataset is not stated.

### 2.3.3. Solar Flux

In contrast to the solar zenith angle, the extraterrestrial radiation only varies seasonally, and not spatially. We used the data provided within the products to obtain a full annual cycle (one value per day of the year) per band, interpolating over days without observations.

## 3. Results and Discussion

### 3.1. Pixel Positioning: Geo Versus Tie-Point

In this section, we discuss the actual field of view of the OLCI instrument, depending on the chosen coordinates for extraction: per pixel geo-coordinates or tie-point. The distances discussed here relate to single pixels, and they have to be extended proportionally in the case of 3-by-3 or other spatial aggregates. Two distance benchmarks are used in this section: 212 m and 700 m. The 212-meter radius circle is the expected pixel center location (see Section 2.2.2). The 700-m-diameter circle is the expected time-series footprint; when all pixel centers are within 212 m from the point of interest, pixel borders are within 362 m (212 m + 300 m/2), resulting in approximately 700 m footprint. The other distances mentioned in this section were measured within QGIS.

SNAP Extract Pixel Values tool conducts extraction exclusively based on geo-coordinates (Figure 2b). Pixel extraction is not trivial, because the coordinates are reported per each pixel, i.e., latitude and longitude are two-dimensional. The only method is thus to calculate the metric distance to the point of interest. In addition, duplicated pixels complicate the task of pixel extraction: duplication of geo-coordinates leads to two (or more) equidistant pixels, forcing the software to make an arbitrary choice. Figure 2b shows that SNAP sometimes chooses pixels incorrectly—the centers of the pixels lay outside the circumference around the location of interest. This happens only in the east–west direction, suggesting that it can be attributed to the degradation of the instrument resolution at the swath edge. As a result, the footprint becomes wider: from the theoretical 700-m-diameter circle to an ellipse with the 700 m south–north minor axis and the 1000 m east-west major axis. Yet, for time-series applications, this is the most accurate data one can get, because it is based on geo-coordinates. We suggest to do manual control and remove the pixels outside the circumference (red vertical threshold line in Figure 3).

An alternative option is to conduct the pixel extraction based on tie-point coordinates (Figure 2c). Notice, however, that this is an approximation, because TP coordinates do not have parallax correction, introducing even more uncertainty at high altitudes. The effective footprint was even smaller—a 550 m diameter circle. However, the true (geo-coordinates based) footprint was way larger and did not have a regular shape (Figure 2d), which is especially noticeable for the Sentinel-3B spacecraft. This fact imposes additional requirements on the homogeneity of the area of interest from 700 m (theoretical) to 2700 m (actual). Given that the average agricultural field area in the European Union is 16.6 ha [60], Sentinel-3 with its 700 m (49 ha) or 2700 m (729 ha) resolution is unsuitable for agricultural area mapping in the EU. Since the GEE product also uses tie-point coordinates (Figure 2e), one might expect the same 2700 m actual footprint (Figure 2d). GEE extraction does not suffer from outliers, but the distance threshold should be increased from 212 m to 225 m, because the pixel size of the GEE product is 318 m (Figure 3). Furthermore, a comparison of tiles c and e of Figure 2 shows that GEE reprojected the images: GEE pixels are strictly north-oriented, while DHUS ones were tilted 15° clockwise for our study site.

Overall, if the area of interest is homogeneous within the 700-m-diameter circle, the only option is to use geo-coordinates (DHUS product) and the time-consuming "data-to-code" approach. At the same time, if the assumption of homogeneity can be extended to the 2700-m-diameter area, GEE products can be a choice.

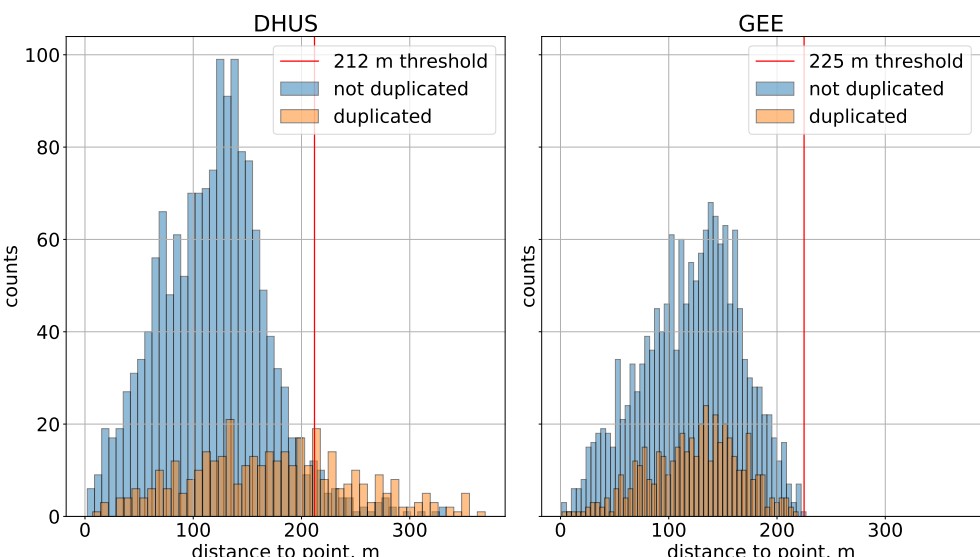

**Figure 3.** Distance from the extracted pixels to the point of interest. The red line denotes the theoretically acceptable threshold. See Section 3.1.

### 3.2. GEE versus DHUS: Radiance Difference

In this section, we compare radiance values reported in the GEE against the genuine DHUS. We matched 1146 products by the full name, which means that their pixel radiances should be identical. The other 741 were matched by the short name, which does not uniquely identify the image (Listing 1), but still helped us to match them unambiguously. Even after the short name matching, a number of products remained unpaired: 21 DHUS products did not have a GEE counterpart and 158 GEE products vice versa (Table 1).

The root mean squared error (RMSE) between GEE and Copernicus OLCI band radiance for single pixel time series is presented in Figure 4. This test quantifies the uncertainty of GEE against properly (per pixel) geo-located pixels for our study site. The users who gap-fill the offline products with the GEE products may expect similar disagreement in radiance. The RMSE did not correlate with the distance to the location of interest, the pixel quality flags, or with anything else that can be detected without direct radiance comparison. It also did not differ for full-name or short-name matching; therefore, the rest of the comparisons were made on all 1887 images together.

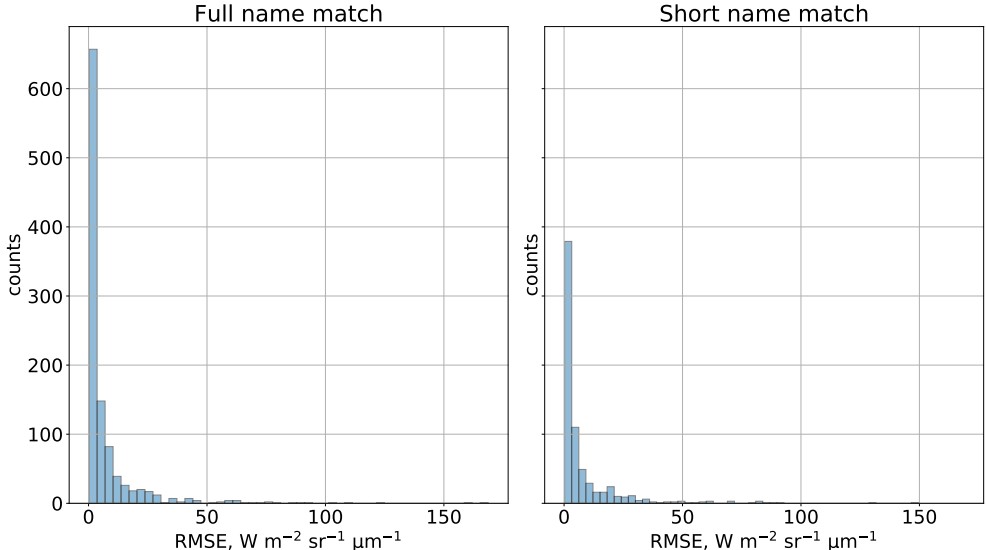

**Figure 4.** Root mean squared error (RMSE) between GEE and Copernicus OLCI products with the same name.

At the next step, we quantified the uncertainty in radiance after spatial averaging of 3-by-3 pixels, using tie-point coordinates (Figure 5). We show the data as relative RMSE (rRMSE) to normalize for systematic difference among bands, and thus facilitate the comparison of bands. We use a (somewhat subjective) threshold of 5% rRMSE as acceptable. As expected, tie-point-based DHUS extraction showed closer values to GEE (which is also tie-point based), especially after the aggregation; however, the values were still not identical.

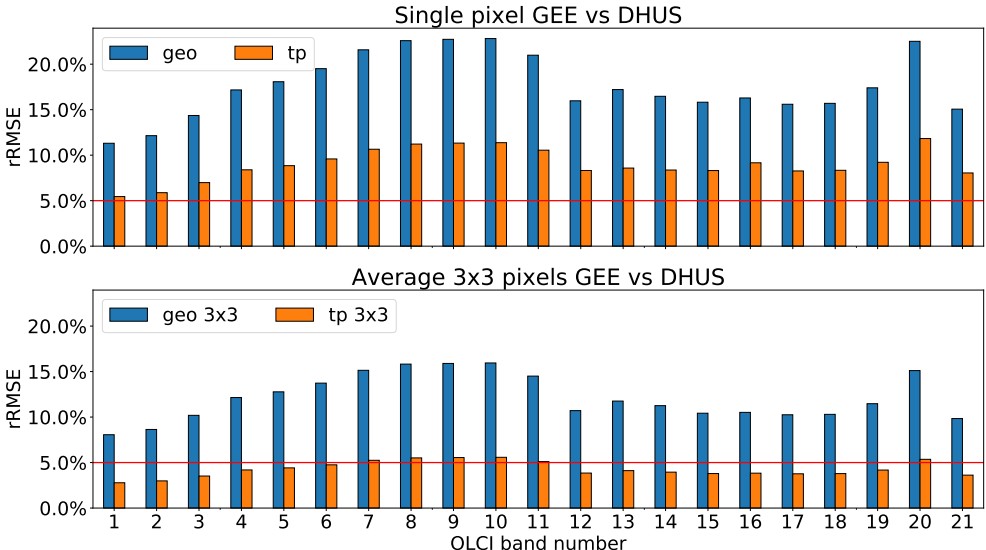

**Figure 5.** Sentinel-3 OLCI from GEE and DHUS. Note that the data are supposed to be identical (rRMSE = 0). The red line denotes the subjective acceptance threshold of 5% rRMSE.

Having discovered that pixel values deviate from the original data, we tried to understand what the problem was—incorrect export or data transformation. We checked our script on MODIS (Figure A1) and Sentinel-2 (Figure A2) datasets and confirmed that for these datasets the script worked correctly and resulted in identical pixel values.

During the search for the data transformation, we found that GEE does reproject the images. We selected 33 products with the highest RMSE, reprojected them in SNAP to the GEE-corresponding CRS, and compared them again. The reprojected products matched in shape, pixel x and y coordinates matched $+/-1$ pixel index, but the RMSE of radiance remained unacceptably high. Changing the SNAP interpolation technique from nearest to bicubic or bilinear did not change the result either.

In conclusion, the tie-point nature of GEE OLCI data resulted in a 10–20% error in radiance for our study site. The cause is the erroneous pixel positioning. The reprojection done by the GEE team during the image ingestion is the most likely cause of the residual 5% rRMSE. We were able to reproduce the image dimensions but not the radiance values with the SNAP reprojection tool. Gridding artifacts, described by Gomez-Chova et al. [18] for MERIS, might have played a role as well.

### 3.3. GEE Augmentation

This section shows the accuracy of the proposed estimation of the geometric and meteorological data missing in the GEE product, but present in the original DHUS. An important note regarding near-real-time (NR) products is that their meteodata differs from non-time-critical (NT) products. In this comparison (and the whole study), we used only NT products.

#### 3.3.1. Angles

The calculation of solar angles from latitude, longitude, and time is straightforward. However, due to varying orbits, wide swath, and satellite inclination, the observation

angles calculation is not that easy. The repeat cycle of 27 days suffices to download all unique relative orbits and use those values to complete the metadata. For NL-Spe, we had 22 frequent orbits and 11 orbits that were encountered only once (Table 2). The latter belongs to Sentinel-3B at the beginning and the end of the tandem phase. For those 11 products (and other 15 observed during the drift of satellites) our method does not work. The accuracy of angles retrieval from orbit numbers is presented in Figure 6. The outliers belong to Sentinel-3B during its fast drifting phase from tandem (from 26 October 2018 to 18 November 2018).

Despite the fact that the relative orbit number is reported in the full name, GEE products always have the relative_orbit_num property. Therefore, it is possible to restore angles even for those GEE images that do not contain the PRODUCT_ID property, i.e. the full 2017 year collection.

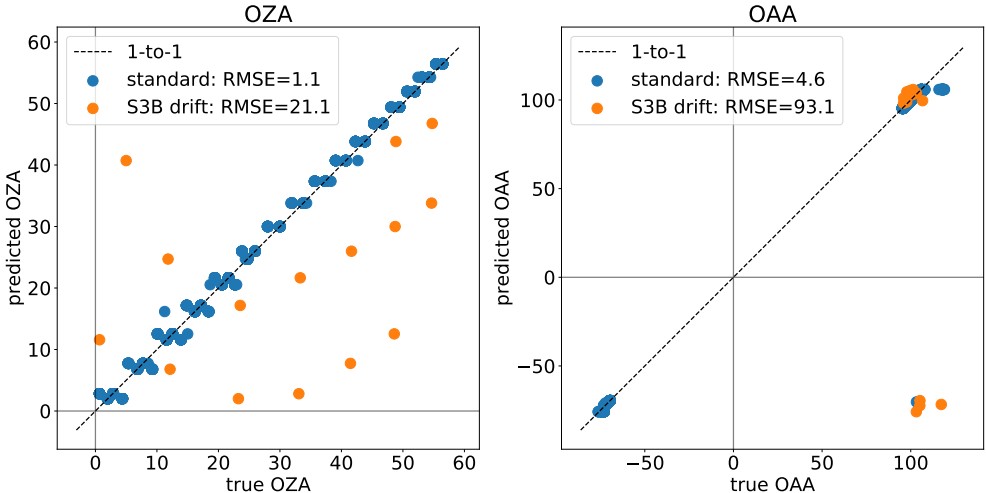

**Figure 6.** Performanceof the estimation of observation zenith (OZA) and azimuth (OAA) angles for GEE OLCI product. Blue dots correspond to the nominal orbits, and orange dots correspond to the orbits taken by Sentinel-3B spacecraft during the drift phase (from 26 October 2018 to 18 November 2018).

3.3.2. Meteorological Data

The meteorological data, missing in the GEE product, can be retrieved from the CAMS dataset. We linearly interpolated the 3-hourly data to the time of the satellite overpass. Figure 7 shows that CAMS-derived values are in a good agreement with the values reported in the original products. Aerosol optical thickness at 550 nm (AOT) is another variable that can be acquired from the CAMS dataset and used for atmospheric correction. However, AOT is not included in the original OLCI product.

Figure 8 demonstrates that the interpolated CAMS values at pressure levels can be used to augment GEE products. However, the RMSE increases at lower pressures (levels 20–25).

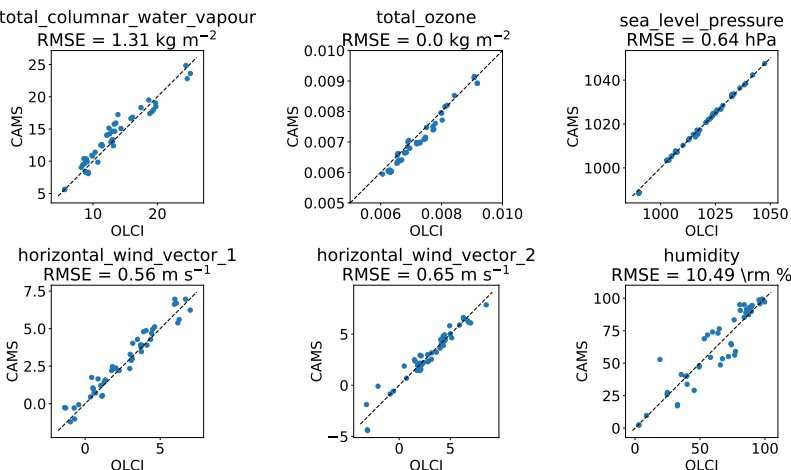

**Figure 7.** Comparison of CAMS-derived (y-axis) and in-product (x-axis) meteorological data.

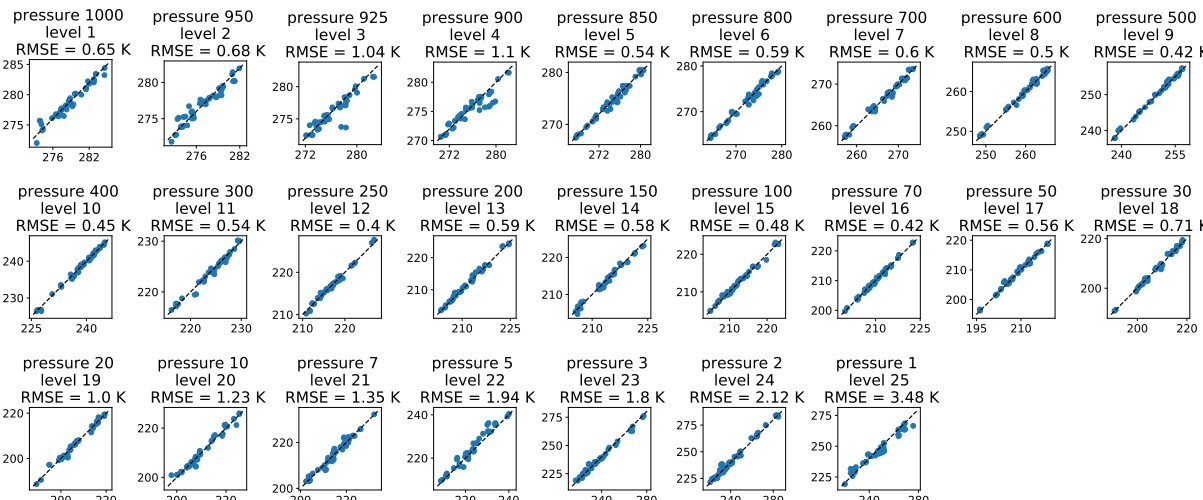

**Figure 8.** Comparison of CAMS (*y*-axis) and in-product (*x*-axis) meteorological data—temperature profile at pressure levels.

### 3.3.3. Solar Flux

Extraterrestrial (TOA) solar irradiance data, missing in the GEE product, can be taken from the mean annual cycle (see Section 2.3.3). We were expecting the estimated value to match perfectly with the in-product value. However, Figure 9 shows that retrieving the solar flux without considering the in-field-of-view pixel position (instrument pixel, provided by the detector_index band) yields large uncertainties in solar irradiance at some bands (especially the first OLCI band Oa01) where local variability with wavelength is the highest. These uncertainties will directly propagate into the conversion of radiances into reflectance, as the presence of per-pixel solar flux in DHUS products is precisely meant for that. Due to cross-FOV central wavelength variations, in-band radiance values can vary by up to 4% within the FOV as demonstrated by Lamquin et al. [61].

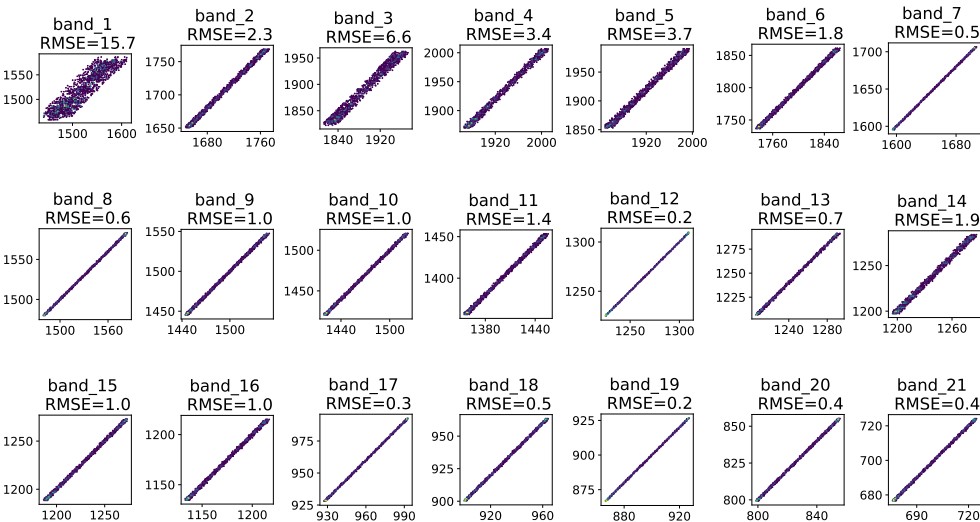

**Figure 9.** Comparison of the estimated (*y*-axis) and in-product (*x*-axis) extraterrestrial (top of atmosphere) solar flux, all in W m$^{-2}$ sr$^{-1}$ μm$^{-1}$.

## 4. Conclusions

We quantified the uncertainty in the GEE OLCI dataset and proposed a method to augment the dataset with meteorological and geometric data distributed with the original ESA products. We expect this to be useful for scientists working with per-pixel time-series, the acquisition of which is complicated by the fact that half of the products are offline in the long-term archive (LTA), and no more than 20 products per day can be requested. We discovered that GEE transformed the original data (rounded values and, probably, conducted reprojection), which led to the deviation of pixel values from the original data of 15 W m$^{-2}$ sr$^{-1}$ μm$^{-1}$ (average RMSE) per band. At least 1 W m$^{-2}$ sr$^{-1}$ μm$^{-1}$ of RMSE can be attributed to the rounding: GEE OLCI values are stored as integers but the reported scaling factors, which should convert them to floating point numbers, are wrong and should not be used. Unfortunately, we could not find any other way of detecting the deviating pixels (by a quality flag or a metadata property), besides the direct comparison of radiance values. Another discrepancy in the GEE ingestion of OLCI products is the usage of tie-point coordinates. Those coordinates lack the parallax correction, which results in larger footprints (4-times larger for our study site: 2700 m vs. 700 m). This imposes the homogeneity constraint on the area of interest, which would increase further in high altitudes.

Suppose the error level of GEE is not acceptable or the area of interest is not homogeneous at 2700 m diameter (9-by-9 pixels). In this case, we propose using the LAADS DAAC archive, which has all genuine products online. However, even with the original products there are challenges in per-pixel time-series acquisition, originating from the absence of fixed tiles and the presence of duplicated pixels (23% of any OLCI image). These require additional distance control and impose the homogeneity constraint of at least 700 m diameter. Despite all the difficulties, pixel extraction from the original OLCI products with SNAP Extract Pixel Values tool (geo-coordinates-based) produces the best possible OLCI-radiance time-series one could get.

We expect that this article will encourage vigilance in using third-party datasets, and will stimulate dataset providers to meticulously describe the data transformation they conducted for the benefit of reproducible science.

**Author Contributions:** Conceptualization, E.P., P.Y., and C.v.d.T.; methodology, E.P.; software, E.P.; validation, E.P.; formal analysis, E.P.; investigation, E.P.; resources, E.P.; data curation, E.P.; writing—original draft preparation, E.P., P.Y., and C.v.d.T.; writing—review and editing, E.P., and C.v.d.T.; visualization, E.P.; supervision, C.v.d.T.; project administration, C.v.d.T.; funding acquisition, C.v.d.T. All authors have read and agreed to the published version of the manuscript.

**Funding:** The project has received funding from the European Union's Horizon 2020 research and innovation programme under the Marie Sklodowska-Curie grant agreement No 721995. Peiqi Yang was supported by the Netherlands Organization for Scientific Research, grant ALWGO.2017.018.

**Institutional Review Board Statement:** Not applicable.

**Informed Consent Statement:** Not applicable.

**Data Availability Statement:** The data presented in this study are openly available in DANS repository at https://doi.org/10.17026/dans-xb8-efke. The code for the original DHUS Sentinel-3 [7] and Sentinel-2 [62] data download and extraction used in this study is available at https://github.com/Prikaziuk/S3_loader. The script used for GEE Sentinel-3 OLCI [53] data extraction is available at https://code.earthengine.google.com/61fe01512385e06b5bc3f65f78bef692?noload=true. The script used for GEE Sentinel-2 level-2 [63] data extraction is available at https://code.earthengine.google.com/abeda236ca3783dd532c44e2fc7c6270?noload=true. The script used for GEE MCD43A4: MODIS Nadir BRDF-Adjusted reflectance [64,65] data extraction is available at https://code.earthengine.google.com/2f4be36dff6109058b6309d9aa9e983c?noload=true. The original MCD43A4 reflectance https://doi.org/10.5067/MODIS/MCD43A4.006 was downloaded with AppEEARS [66]. Copernicus Atmosphere Monitoring Service (CAMS) dataset [58] was downloaded manually.

**Acknowledgments:** The authors thank Ludovic Bourg (ACRI-ST) from the Sentinel-3 Mission Performance Centre for technical clarifications, Timofei Bondarev for Python code review, Marco Peters and his colleagues from STEP forum (Brockmann Consult) for the answers, and Jae Evans from University of Twente Language Centre for help with English. This research was supported by the Action CA17134 SENSECO (Optical synergies for spatiotemporal sensing of scalable ecophysiological traits) funded by COST (European Cooperation in Science and Technology, www.cost.eu, accessed on 3 March 2021).

**Conflicts of Interest:** The authors declare no conflict of interest.

## Appendix A. Sentinel-3 Products Availability

**Table A1.** Sentinel-3 products availability from the Since date to 23 September 2020: Level 1—top of atmosphere, level 2— surface.

| Instrument | Product Type | Content | Resolution, m | Since | Structure | # Images | Offline | Size, MB |
|---|---|---|---|---|---|---|---|---|
| OLCI | OL_1_EFR | 21 bands | 300 | 2016-04-26 | frame | 2164 | 901 | 620 |
| | OL_1_ERR | 21 bands | 1200 | 2016-04-26 | stripe | 2164 | 901 | 700 |
| | OL_2_LFR | 2 indices, 2 TOC red bands | 300 | 2016-04-26 | frame | 2164 | 957 | 120 |
| | OL_2_LRR | 2 indices, 2 TOC red bands | 1200 | 2016-04-26 | stripe | 2164 | 957 | 170 |
| SLSTR | SL_1_RBT | 24 radiance/10 BT | 500/1000 | 2016-04-19 | frame | 4372 | 2715 | 430 |
| | SL_2_LST | 2 indices, LST, masks | 1000 | 2016-04-19 | stripe | 4548 | 2842 | 60 |
| | SL_2_FRP | ? | ? | ? | ? | | | |
| Synergy | SY_2_SYN | 26 bands, AOT550 and exponent | 300 | 2018-10-08 | frame | 1146 | 3 | 400 |
| | SY_2_VGP | 4 bands, atmosphere | 1000 | 2018-10-09 | stripe | 1133 | 2 | 50 |
| | SY_2_VG1 | 4 bands, NDVI, atmosphere | 1000 | 2018-10-04 | tile | 1374 | 3 | 120 |
| | SY_2_V10 | 4 bands, NDVI, atmosphere | 1000 | 2018-09-22 | tile | 136 | 0 | 250 |
| SRAL | SR_1_SRA | ? | 300 × 1640 | 2016-03-01 | stripe | 696 | 3 | 52 |
| | SR_1_SRA_A | ? | 300 × 1640 | 2016-04-07 | line | 351 | 99 | 2300 |
| | SR_1_SRA_BS | ? | 300 × 1640 | 2016-04-07 | line | 351 | 99 | 1700 |
| | SR_2_LAN | ? | 300 × 1640 | 2016-03-03 | stripe | 713 | 1 | 100 |
| | SR_2_WAT | ? | ? | ? | ? | | | |

## Appendix B. Performance of the Extraction Script on Other Google Earth Engine Datasets

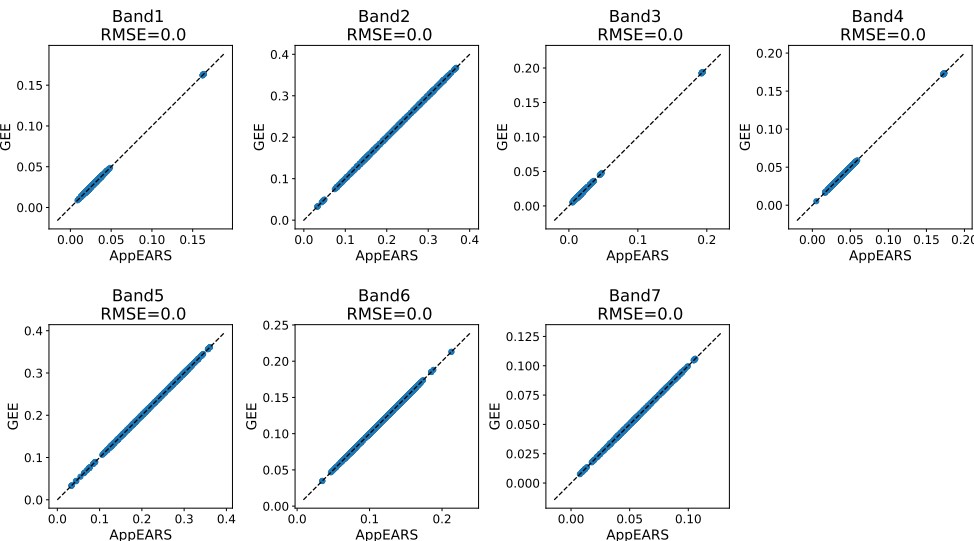

**Figure A1.** MCD43A4-006 [64] single-pixel reflectance time-series from GEE (*y*-axis [65]) and AppEARS (*x*-axis [66]) datasets. The nearest pixel to Speulderbos forest site (NL-Spe, 52.251185°N, 5.690051°E) for the period from 1 January 2016 to 1 October 2020 (1736 products).

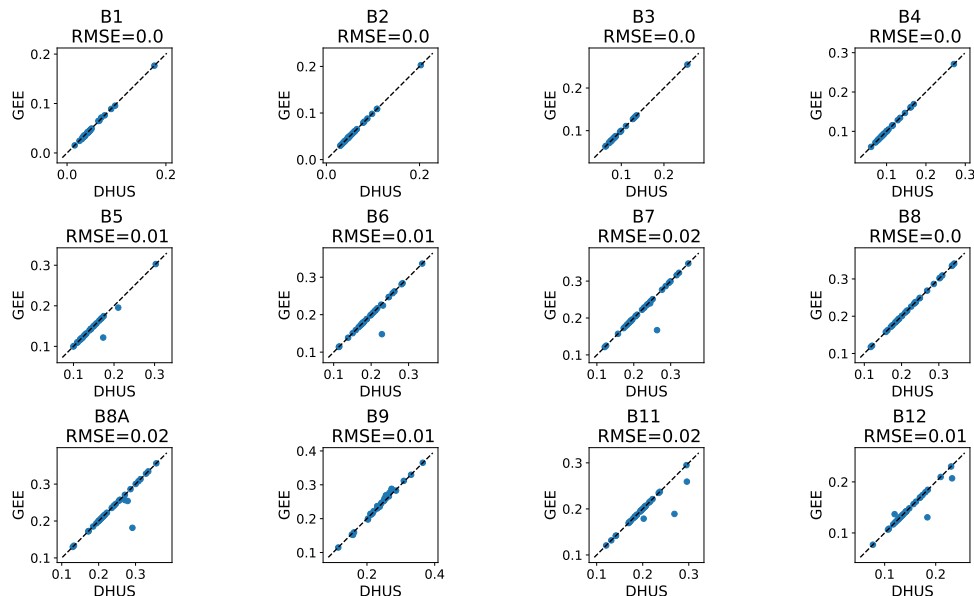

**Figure A2.** Sentinel-2 level-2 [62] single-pixel reflectance time-series from GEE (*y*-axis [63]) and DHUS (*x*-axis [54]) datasets, both resampled to 20 m. In spite of the outliers in bands with native 20 m resolution (B5, B6, B7, B8a, B11, B12), probably caused by GEE pixel weighting during the resampling [67], all pixels demonstrate the expected behavior. The nearest pixel to the Sudan agricultural field site (14.40551°N, 33.39137°E) for the period from 1 July 2019 to 31 March 2020 (54 products).

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
