# Peer review of "Google Earth Engine Sentinel-3 OLCI Level-1 Dataset Deviates from the Original Data: Causes and Consequences"

_remotesensing, doi:10.3390/rs13061098_

Round 1

Reviewer 1 Report

Accept with minor corrections

Revision of the manuscript entitled “Hidden Processing of Sentinel-3 OLCI Level-1 Data by Google Earth Engine Breaks Traceability"

General comments

The study's objective quantified the uncertainty in the GEE OLCI dataset and proposed a method to augment the dataset with meteorological and geometric data from the ESA products. I found some merits in the both methodology and results. In my opinion, this paper has a good potential to be published in the journal. In summary, the data analysis was not complicated and presented well-written results. The authors provided significant limitation and challenges in the manuscript relevant for further studies. However, to holds the reader's attention, I would encourage them to think about the following main concerns and address them in the manuscript. Also slight English language editing is required. Overall, the manuscript merit acceptance for publication with minor revision (see detailed comments below).

a)       Title:

Overall, the title is appropriate for the paper.  

b)       Abstract:

Overall, The abstract is well written and informative. However, the authors need to include quantitative results.

Minor comments

Line 8: Sentinel Application 197 Platform (SNAP)

 Line 8: insert “period” and delete “and” to start a new sentence; delete “case of*. We … original data. In the SNAP Pixel Extraction Tool, it is used to conduct an additional distance check and exclude pixels that lay further than 212 m from the point of interest.

Line 10: delete “the application! And replace with “applying”

Line 11: insert “a” to read “… a 700-meter-…”

Line 13: insert “the” to read “…in the radiance of 10% …”

Line 16: delete “a” to read “… and use …”

Line 18: insert “the” to read “in GEE OLCI dataset”

c)        Keywords:

The authors provided adequate keywords. The keywords are appropriate and would help the article search results in the future or increase the article's visibility to a large audience. It is within the journal’s specified number. However the authors may delete the following and maintain the abbreviations instead.

Google Earth Engine; Ocean and Land Color Instrument; Copernicus Atmo22 sphere Monitoring Service

d)       Introduction

The introduction is well written. However, I think slight language editing is needed to improve the overall of the manuscript.

Minor comments

Line 31: insert “comma” ; insert “of” to read “downloads per user and, a maximum of 20 requests per”

Line 34 and 40: delete “own”

Line 51: replace “been exploring” with “explored”

Line 53: hyphenate real and world to read “real-world”. Insert comma after data

Line 55: insert comma after 2017; delete “collection of”

Line 58: replace up to with “until”; insert “the” before google-earth.

Line 60: insert comma to read “ In this article, we …”

Line 67: insert “the” before Sentinel; replace “with emphasis on “ with “emphazing”

e)       Materials and Method

Overall, the methodology used in this section is acceptable. The research procedures and techniques used for the review are standard for this scientific research and are reproducible. A statement to make state the limitation(s) in this study may be useful. The results are clear, well presented, and provide a basis for developing recommendations to assist policymakers and other stakeholders. However, I think slight language editing is needed to improve the overall of the manuscript.

Minor comments

Line 77: delete “of” to read “OLCI on board Sentinel-3”

Line 79: insert “the” before visible …”

Line 89: insert “the” to read “In the case of OLCI, …”

Line 102: check spelling of “ware”. It should be “were”

Line 116: insert “the” to read “the phenological stage”

Line 122: insert “the” to read “used the time series of OLCI”

Line 127: delete “for” in the sentence

Line 143: insert “the” to start the sentence “The coordinates …”

Line 148: replace “comma” with “semi-colon” to read “(not only cloud-free); thus”

Line 167: delete “of” and “nature” to read “… which are different.”

Line 168: insert “the” to read “the satellite  ...”

Line 176: insert “the” to read “in the GEE OLCI”

Line 177: delete “s” NetCDF.”

Ine 183; insert comma to read “… system (CRS), and …”

Line 184: insert “an” to read “has an … “; delete “clearly” and start the sentence as “The …”

Line 199: insert “the” to read “from the product ..”

Line 206: use one “perod” or “time” not both.

Line 211: insert “the” to read “(the equivalent …”

Line 217-218: use appropriate reference style: https://mt1.google.com/vt/lyrs=s&x={x}& 218 y={y}&z={z}).

Line 218: insert “the” to read “with the distance …”

Line 224: insert “comma” to read “For DHUS, …”

Line 225: insert “comma”; delete “in the”; replace semi-colon with period to read “For GEE, the number of products filtered by geo-region collection was 6795. However, ..”

Line 227: delete “at all”

Line 229: insert comma after metricsLine 232: delete “In fact,” to read 2The …”

Line 235: replace comma with period. Replace “but” with Still” to read “… (2018-12-11). Still, the”

Line 238-239: insert “the” to read “ the processing unit”

Listing 1. – The text of the table is unreadable. It is difficult to understand the text

Line 258: replace comma with period to read “and time. However, “

Line 266: insert “the” to read “applying the atmospheric correction. We downloaded the data from the near-real time”Line 282: insert “the” to read “In contrast to the solar zenith”

f)        Results and Discussion

The results are clear, well presented, and provide a basis for developing recommendations to assist policymakers and other stakeholders. This section is overall well complemented with few figures that help to visualize the results.

However, I think slight language editing is needed to improve the overall of the manuscript.

Minor comments

Line 291: insert “the” to read “in the case of “

Line 300: insert “the” to read “at the swath edge”

Line 302: insert “the” to read “along the east-west”

Line 308: replace “which can bring” with “bringing”

Line 311: insert “a” to read “a 550-meter”

Line 317L: insert “the” to read “in the EU”.

Line 320: insert “the” to read “size of the GEE product”

Line 322: replace “whilst” with “while”; replace “were” with “was”

Line 327: insert “a” and “the” ; delete “alternative” to read “a 2700-meter-diameter area, the GEE product can be a choice.

Line 330: 1146???? Why start a sentence with a figure?

Line 331: 741???

Line 338: replace “to” with “do”

Lin e340: replace “done” with “made”

Line 341: insert comma to read “As the next step, we …”

Line 350: confirmed

Line 354: insert them to read “compared them again”

Line 358: insert “a” to read “in a 10-20% “

Line 360: insert “the GEE”

Line 362: insert “the” to read “with the SNAP”

Line 372: delete “with” to read “The repeat cycle of 27 days suffices”

Line 375: delete “at”to read “and end of “; add “s” in products; insert “the” to read “11 products (and the other”

Line 376: insert “the” to read “during the drift of”

LInee 378: replace “are with “is”

Line 381: delete comma and start sentence to read “property. Therefore”

Line 386: delete “that can be”; insert “the” to read “from the CAMS”

Line 394: Oa01???

g)       Abbreviations:

All abbreviations are defined at first except Sentinel Application 197 Platform (SNAP in abstract

h)        Conclusion

This section is well written. The conclusions were based on the findings of the results and logically stated. The authors provided limitations and challenge faced during the study (Line 413-418) which is relevant for future studies. However, I think slight language editing is needed to improve the overall of the manuscript.

Minor comments

Line 401: insert “the” to read “in the GEE OLCI”

Line 405: replace “long-therm” with “long-term”

Line 408: replace “were not able to” to “could not”

Line 408-410: “Unfortunately, we were not able to find any other way 409 of detecting the deviating pixels (by a quality flag or a metadata property), besides direct comparison of radiance values.” I am sure what the authors seem to communicate to their audience with this sentence. I am curious to understand why? Was the problem a methodological problem or based on the data? Was this stated in the methodology and results sections? I need more clarification for this statement.

Line 410-411: “In addition, scaling factors reported in GEE should not be used.” This statements is unclear. I think it needs editing.

Line 408-411: The sentence structure is not readable especially linking it to the Line 411-413.

Line 412: insert “the” to read “using the LAADS DAAC”

Line 411-413: This reads better “Suppose the error level of GEE is not acceptable or the area of interest is not homogeneous at 2700 m diameter (9-by-9 pixels). In that case, we propose using the LAADS DAAC archive, which has all genuine products online.”

Line 417: insert “the” to read “further in the case”

i)        Reference

References used are appropriate and relevant. Make sure URL used in the manuscript follows MDPI format or reference style

j)         Data availability statement and code used

The authors should provide a section for this. If possible include links to access the data and code

Recommendation

Accept with minor revision

Author Response

Dear reviewer, thank you for your time and comments. Please, find our response attached.

The authors

Reviewer 2 Report

This paper by Prikaziuk et al. concerns some discrepancies appearing in Sentinel-3 data when the Google Earth Engine is used for data acquisition. Additionally, examines differences in pixel geolocation that may appear when different coordinates, included in the original products, are used. The subject of the paper is interesting, since Setinel-3 data are promising for earth observation applications, full Level-2 products are not available yet, even after 5 years after the first Sentinel-3 launching and potential end users may not be aware of these problems. In my opinion, an advantage of the work done is the use of free software (Google Earth Engine, SNAP, QGIS) which are gaining attention in the remote sensing community. However, since the paper is a technical note, more methodological details – especially concerning the use of SNAP – should be given, for the potential Sentinel-3 end users to be able to reproduce and resolve the problems described.  Additionally, in some cases, problem description and the proposed solutions are described in a rather difficult to follow way (especially Figure 2) and a clearer description would add up to the paper readability. Finally, even though it is mentioned in the text, it should be stressed more clearly that the described problems are important especially when time series for only one pixel are needed (which may often be the case for studies combining satellite and eddy tower data, since many European eddy sites are surprisingly small) and less important for large and homogenous sites. Overall, the paper will be suitable for publication after some revisions on the following issues:

L44. the example → an example

L98. Revisit time: to my knowledge, with one Sentinel-3 revisit time is 1 to 2 days and with 2 satellites becomes daily, with many days having images from both Sentinels (A and B). This may be easily seen with a simple query in the Copernicus DHUS.

L150. It is not clear what the authors mean by “23% of any image”. Are the 23% of the pixels duplicated in all Sentinel-3 images?

Figure 1. For a reader familiar with SNAP the figure may be rather explanatory. However, non-SNAP users may have difficulties understanding the information of the figure and the legend does not help either. For example, what is TP_latitude and its difference from latitude (both shown in the figure)? Even though this is explained in the text, the details are given after Figure 1 and the presence of the two TP_ columns in the Figure may be confusing.

L183-184. “and more often than not has x-size of 4866” please rephrase.

L202-205. It would be useful for the reader to explain in detail how pixel extraction based on geo-coordinates or tie-point coordinates was made using SNAP. Do the authors mean using the option “Read Sentinel-3 OLCI products with per-pixel geo-coding instead of using tie-points” found in SNAP Tools/Options/S3TBX (see image in the pdf file) or using the TP_ coordinates that are included in the data after data extraction? If the former is the case then the different approaches do not concern data extraction but product reading. Please clarify.

L214-216. I suppose that the 212 m radius circle is what is indicated as “expected pixel center location” in Figure 2. If that is correct, I think it would add clarity if it was clearly stated both here and in Figure 2.

Listing 1. The names of the two products are not fully shown (see authors remark in the legend).

L241-242. It is not clear what is not present in DHUS: products from Svalbard Satellite Core Ground Station (SVL) in general, or the specific 211 GEE products?

L245. where → were??

L292. “By default SNAP Extract pixel tool uses geo-coordinates”. There is no relevant option in SNAP Extract pixel tool. Do the authors mean the reading option mentioned above (L202-205)? If that is the case, then the SNAP default is with that option unchecked, which means that products are read based on tie-points. Please clarify.

L301-302. How was the theoretical 700 m diameter was estimated?

L309-311. Why 7x7 pixel extraction? A regular user would probably extract only the one pixel containing the coordinates of interest. Please explain.

L315-316. “Given that the average agricultural field area in the 316 European Union is 16 ha” a reference would be nice.

L321-323. I suppose that tilting is concluded by the shape formed by the points in Fig c and e. However, showing two images for the same date from DHUS and GEE would probably show tilting more clearly.

Figure 2. What do the authors mean by “expected pixel location”? How was the effective footprint calculated? Is it the line surrounding pixel centers at a distance of 212 m to the closest one? Do A and B in the legends correspond to Sentinel-3A and 3B satellites? Please give more details both in MS text and Figure legend and make the legend more informative.

L392-399. It is not clear to me what is compared with what. Please clarify the procedure.

L398. What does E0 stand for?

Figure 9. Legend is a bit confusing to me: what is interpolated and how? Please clarify. x-axis labels are overlapping in some graphs.

Conclusions. After clarifying the default SNAP extract procedure (see comments for L292 and L202-205), I think it would be useful for the end user a clear suggestion on which procedure is more accurate: based on geo-coordinates or tie-point coordinates? Please include that information both here and in the abstract.

Author Response

Dear reviewer, thank you for your time and feedback. Please, find attached the answers to your comments.

Best regards,

The authors

Reviewer 3 Report

Dear authors,

The reviewed technical report solves actual problem that might be worth broadly discussed and requires deeper investigation and co-operation with GE developers. I think researches know GE imagery limits and do not use mindless these GE “translated” satellite images. However, each research study helps Google developers make the cloud computing technology more precise and widespread. Cloud-based data computing is close future. Generally, the report brought interesting findings.

Academic debate maybe might initiate processing of time-series of satellite imagery in Google Earth Engine (GEE) cloud platform and the same time-series in desktop GIS, e.g. QGIS and the comparison of geometry shifts or possible errors. However, it would be possible only in case when Google Earth Engine imagery processing algorithms are known and reproducible. Therefore, I suggested to change title, for instance: "Discrepancies of Sentinel-3 OLCI Level-1 Data from Google Earth Engine affecting satellite images time-series processing" or to other one more precisely specifying the GEE issue – what is the real problem of the GEE satellite imagery usage. Because a word “hidden” opens a door to a broad academic debate and speculations that might this article start.

Some formal mistakes considering references were found –hypertext links instead numbered citations. It depends on the journal policy to be accept or not. I suggested to use common MDPI reference style for online materials.

Please find detailed comments in attached PDF file: remotesensing-1119133-peer-review-v1.pdf

and bellow arranged in lines respectively:

  • Title is a little difficult to read and understand what does it mean "hidden" try to reword to be more precise e.g. "Discrepancies of Sentinel-3 OLCI Level-1 Data from Google Earth Engine affecting satellite images time-series processing"
  • Lines 56, 59, 100, 170, 174, 193, 196-197, 209, an other: please cite hypertext links using the MDPI reference style of online documents: https://developers.google.com/earth-engine/issues; https://groups.google.com/g/google-earth-engine-developers/c/cssvsuITy30/m/pXCj8KzoBQAJ; https://s3tandem.eu/ ….
  • 60-64: move at the end of the assumed aim (1), or use common numbering: 1. xx; 2. xxx ...
  • 67, 70: of the Sentinel-3 OLCI product; sections describe
  • 87, 113, 177 and Footnote 1: explain shortcut “TOA” “Top-of-Atmosphere “
  • 179-180: Refer this finding in supplementary material, appendix or citation.
  • Figure 2e: Comment this value 675 m in results, I could not find it.
  • 356-357: Position error is known issue of NN method and also Gomez-Chova concluded: "For instance, the commonly used nearest neighbour interpolation will provide a high interpolation error. More appropriate interpolation and/or filtering techniques might be used to mitigate such interpolation errors at the expense of a lower spatial resolution." therefore it is interesting finding that the usage of other methods brought the similar results.
  • 363: Did you mean "gridding artifacts", try to explain deeper the problematic. Reprojection can be caused by multiply factors according - Gomez-Chova referred works. In this study it seems that only GEE developers might know the reason and maybe not. In further study, it might be interesting to collect outputs of previous studies and using multivariate correlation to assess the significance of correlated variables / factors causing geometry errors - if there exist any dependency, maybe not.
  • Figure A1: Drusch et al. I must be cited in the reference list.

regards,

Reviewer

Author Response

(The authors gave the same response as above.)
